# Life events and adolescent depressive symptoms: Protective factors associated with resilience

Kristin Gärtner Askeland[1]*, Tormod Bøe[1,2], Kyrre Breivik[1], Annette M. La Greca[3], Børge Sivertsen[4,5,6], Mari Hysing[1,2]

**1** Regional Centre for Child and Youth Mental Health and Child Welfare, NORCE Norwegian Research Centre, Bergen, Norway, **2** Department of Psychosocial Science, Faculty of Psychology, University of Bergen, Bergen, Norway, **3** Department of Psychology, University of Miami, Miami, Florida, United States of America, **4** Department of Health Promotion, Norwegian Institute of Public Health, Bergen, Norway, **5** Department of Research & Innovation, Helse Fonna HF, Haugesund, Norway, **6** Department of Mental Health, Norwegian University of Science and Technology, Trondheim, Norway

\* kristin.askeland@norceresearch.no

**Data Availability Statement:** Data cannot be shared publicly due to privacy restrictions in accordance with the ethical approval for the youth@hordaland-survey. Norwegian Health research legislation and the Norwegian Ethics

## Abstract

### Introduction

Depression is a public health concern among youth, and it is pertinent to identify factors that can help prevent development of depressive symptoms in adolescence. This study aimed to investigate the association between negative life events and depressive symptoms among adolescents, and to examine the influence and relative contributions of personal, social and family protective factors related to resilience.

### Methods

Data stem from the cross-sectional youth@hordaland-survey, conducted in Hordaland, Norway. In all, 9,546 adolescents, aged 16–19 years old (52.8% girls) provided self-report information on depressive symptoms, negative life events and protective factors related to resilience.

### Results

Experiencing a higher number of negative life events was related to increases in depressive symptoms, while the potential protective factors goal orientation, self-confidence, social competence, social support, and family cohesion individually were associated with fewer symptoms. Although there were small moderating effects of goal orientation and self-confidence, the results mainly supported a compensatory resilience model. When considering the potential protective factors jointly, only self-confidence and family cohesion were significantly associated with fewer depressive symptoms for both genders, with the addition of social support for girls. There were significant interactions between all the potential protective factors and gender, indicating a greater reduction of depressive symptoms with higher levels of protective factors among girls.

committees require explicit consent from participants in order to transfer health research data outside of Norway. In this specific case, ethics approval is also contingent on storing the research data on secure storage facilities located in our research institution. Data are from the Norwegian youth@hordaland study, owned by NORCE Norwegian Research Centre. The authors did not have special access privileges to data from the survey. Researchers who meet the criteria for access to confidential data can apply for access to data in the same manner as the authors, following the procedures of the Bergen Child Study. For more information, please contact bib@norceresearch.no.

**Funding:** This research was funded by Extrastiftelsen, grant number 2018/FO202170 awarded KGA. URL of Extrastiftelsen: https://www.extrastiftelsen.no/ The funders had no role in study design, data collection and analysis, decision to publish, or preparation of the manuscript.

**Competing interests:** The authors have declared that no competing interests exist.

## Conclusions

Interventions aimed at fostering self-confidence and family cohesion could be effective in preventing depressive symptoms for adolescent boys and girls, regardless of their exposure to negative events. Results further indicate that preventive interventions targeting these potential protective factors could be especially beneficial for adolescent girls.

## Introduction

Depression is of great public health concern [1], with serious consequences both for affected individuals and the wider society [2, 3]. Depressive symptoms increase between childhood and adolescence, especially among girls [4–6]. In adolescence, girls are about twice as likely to be depressed than boys, a gender difference that persists into adulthood both for depressive symptoms and diagnosable disorders [7, 8]. The etiology of depression is complex. In addition to a familial history of depression, exposure to psychological stress is one of the strongest risk factors for depression in adolescence [9, 10]. The risk is greater when adolescents are exposed to multiple stressful events [11]. Given the increase in depressive symptoms in adolescence and the negative consequences, it is important to identify factors that might protect against the development of depressive symptoms in the face of known risk factors.

Resilience refers to the process wherein an individual copes well and has a relatively good outcome, even when exposed to risk factors that may disrupt normal development [12–15]. Positive development is in itself not sufficient to establish that resilience is present [15, 16]; in addition, there must be current or past risk with a known potential to disrupt development [14]. Thus, positive adjustment refers to an outcome of resilience, while resilience in itself is the process of overcoming risk [16]. Further, the process of resilience is complex, and the importance of different protective factors may vary according to the specific risk involved and the outcome being studied [16]. Protective factors related to resilience stem from multiple domains, and may include factors within the person, the family and the broader social environment [13, 17].

Research into potential protective factors (henceforth called protective factors) for adolescents at risk for developing depression have included all these domains, and social support has received particular attention. Studies categorizing adolescents into groups based on outcomes and risk exposure have found that adolescents described as resilient (i.e. who show positive outcomes in the presence of risk) report more positive relationships with their family and more social support [18–21]. A recent meta-analysis concluded that social support from the family is protective against depression; however, the findings are less consistent regarding peer support [22]. Several important personal protective resources have also been identified, including a positive self-concept [19], higher self-esteem [23], optimism and perceived mastery [18], higher personal competence, a structured style [21], and use of active coping strategies [18, 23–25].

Protective factors related to resilience are diverse and often interrelated, and it is important to investigate the concurrent impact of multiple protective factors, as some might only be important when studied alone [26]. For instance, social support no longer discriminated between groups labeled as resilient and vulnerable when self-efficacy and coping strategies were accounted for [23]. Similarly, parental bonding had a moderating effect on the relationship between negative life events and depression only when examined alone, not when cognitive strategies, such as positive reappraisal, were also considered [25]. As many studies focus

solely on protective factors from one domain, or investigate multiple protective factors independently of each other [21, 27], it is important to assess their individual contributions when jointly studied.

Another important consideration is the possibility of gender difference. Girls report more-depressive symptoms [4, 6, 8, 28] and also experiencing more negative life events than boys [29]. Regarding protective factors related to resilience, studies suggest that girls have higher scores on social competence [27, 30–32], while boys have higher scores on self-confidence [27, 30–32] and self-esteem [33]. It is therefore important to investigate whether the influence of protective factors on the association between negative life events and symptoms of depression is gender specific. Some evidence suggests gender differences; specifically, positive peer relationships bufferedthe effects of stress on depression for boys, whereas cohesive family relationships buffered the effects of stress for girls [34, 35]. Still, the majority of studies have not investigated these associations separately for boys and girls, although identifying possible gender differences could be important in order to develop gender-specific preventive interventions.

Two of the most influential theoretical models on resilience are the compensatory model and the protective model [12, 36, 37]. A compensatory model is evident when a factor has a direct influence on the outcome of interest (i.e., a main effect in the analysis), and does not interact with a risk factor in predicting the outcome [12, 36]. Thus, the protective factor applies for all in the same manner, both those exposed and not exposed to the risk. A protective model is evident when the protective factors are especially influential when risk is present. It is identified when the protective factor interacts with the risk factor to predict the outcome [12, 36] and does not influence the outcome to the same degree unless risk is present. Identifying such interactions require large samples, as the statistical power is influenced by the measurement error in both included variables [38]. A central limitation of previous research is the use of samples that might be too small to detect interactions [10, 21, 25].

Based on these considerations, the aim of the present study was to investigate the association between negative life events and symptoms of depression for adolescent boys and girls, and to examine the influence of protective factors associated with resilience. These include goal orientation, self-confidence, social competence, social support and family cohesion. In these analyses, a central aim was to assess whether the results supported a compensatory or protective model of resilience. Lastly, we aimed to investigate the relative contributions of personal, social and family related protective factors on adolescents' symptoms of depression.

## Method

This study is based on the population-based youth@hordaland-survey, conducted in the County of Hordaland, Western Norway in the spring of 2012. All adolescents born from 1993 to 1995 and residing in the county at the time of the survey were invited to participate. The survey consisted of a web-based questionnaire covering information on a range of demographic background variables, lifestyle factors, mental health problems and resilience. Adolescents enrolled in upper secondary education received information about the study and the log on information needed to participate to their school e-mail address. Adolescents not enrolled in school received this information by postal mail to their home address. The schools allocated one school hour (approximately 45 minutes) for completion of the web-based questionnaire. School staff were present at the time of the data collection to ensure confidentiality, and survey staff were available on telephone to answer any questions from teachers or students regarding the survey. In addition, the adolescents could complete the questionnaire at their own convenience throughout the data collection period.

## Ethics

The study was approved by the Regional Committee for Medical and Health Research Ethics (REC) in Western Norway (2011/811/REK vest). In accordance Norwegian regulations, adolescents aged 16 years and older can make decisions regarding their own health (including participation in health studies), and thus gave consent themselves to participate in the current study. The lack of consent from parents/guardians was approved by REC. The present study is a part of a larger project preregistered at Open Science Framework (https://osf.io/kpqwe/).

## Sample

In total, 10 257 adolescents (ages 16 to 19 years) responded to the survey, yielding a participation rate of 53%. For the present study, adolescents with missing information on the variables assessing negative life events were removed from the sample (631 adolescents). To ensure the quality of the data, adolescents also were removed if they reported that they were older when the event happened than their actual age at the time of the survey (71 adolescents). Further, we investigated whether adolescents who had given obviously erroneous answers on other parts of the questionnaire answered these variables differently from the total sample. This resulted in removal of another 9 adolescents.

The final sample size for the present study was 9 546 (93% of the adolescents who responded to the survey). There was a higher proportion of girls in the present sample (53.9%) compared to the total sample (52.6%), while the mean age was 17.4 years in both samples.

## Instruments

**Demographics.** Age and gender of the participants were derived from the personal identification number from the Norwegian National Registry. The adolescents reported their mother's education with the response alternatives: 'primary school', 'secondary school', 'college or university: less than four years', 'college or university: four years or more', and 'don't know'. For the purpose of the present study, the two categories indicating college or university education were combined into 'college or university' regardless of the length of the education.

**Negative life events.** Negative life events were measured by the question: 'Have you ever experienced any of the following events?' followed by the response alternatives: 'death of someone close to you', 'a catastrophe or serious accident', 'violence from a grownup', 'witnessed someone you care about being exposed to violence from a grown up' and 'unwanted sexual actions'. The response alternatives for each event were: 'no, never', 'yes, once', 'yes, some times' and 'yes, several times'. If the adolescent had experienced the death of someone close, they were asked who that person was with the response alternatives: 'parent/guardian', 'sibling', 'grandparent', 'other close person in the family', 'close friend' and 'girlfriend/boyfriend'. Multiple responses were possible.

A variable indicating the total number of negative life events experienced was calculated. For each negative event, a response of 'yes, once', yes, some times', or 'yes, several times' was used to indicate exposure to the negative life event in question. Regarding the death of someone close to you, death of a parent/guardian, sibling, close friend and girlfriend/boyfriend were included as separate negative events, giving a total of eight possible events. Due to the low number of adolescents who reported five or more negative life events, responses of four or more were collapsed into one category, giving the alternatives: 0, 1, 2, 3 and 4 or more.

**Depression.** Symptoms of depression were assessed using the Short Moods and Feelings Questionnaire (SMFQ) [39]. The SMFQ consists of 13 items measuring cognitive and emotional symptoms associated with depression experienced by the adolescent during the past two weeks. The items are answered on a 3-point Likert scale with the response alternatives: 'not

true', 'sometimes true' and 'true'. The SMFQ has shown good psychometric properties in population-based studies [40, 41], and has previously been validated in the sample from youth@-hordaland [28].

**Resilience.** Resilience was assessed using the Resilience Scale for Adolescents (READ) [31]. The READ consists of 28 positively formulated items rated on a 5-point Likert scale ranging from 'totally disagree' (score of 1) to 'totally agree' (score of 5). Higher scores on the READ indicate higher levels of the protective factors associated with resilience. The factor structure and psychometric properties of the READ has previously been tested in the sample from the youth@hordaland [30], suggesting that the items asses five factors: Goal Orientation, Self-Confidence, Social Competence, Social Support, and Family Cohesion. Four of the items in the original READ are not included in these factors (item 4, 9, 12 and 25). Goal Orientation assesses planfulness and organizational skills, while Self-Confidence assesses feeling competent and believing in one's abilities. Social Competence assesses the ease of making new friends and talking to people. Social Support measures having someone who cares, encourages and can help. Family Cohesion assesses support and shared values in the family (for an overview of items, see S1 Table). The READ factors will be referred to as protective factors in the following to ease readability, despite the cross-sectional nature of the data.

## Statistical methods

The SMFQ scores had a right skewed distribution deviating from normality (skewness 1.3, kurtosis 4.3). Nevertheless, as the independent samples t-test is robust to deviations from the assumption of a normal distribution in large samples [42], gender differences in symptoms of depression were investigated by independent samples t-test. Gender differences in age also were investigated by independent samples t-test, while gender differences in maternal education and number of negative life events were investigated by chi-square tests. The association between negative life events and symptoms of depression was investigated in a regression analysis where three models were specified. In Model 1, the number of negative life events was included as a predictor of depressive symptoms. In Model 2, age, gender and maternal education were included, and in Model 3 an interaction term between gender and number of negative life events was included. The results of the analysis including the interaction term was plotted for the different levels of negative life events, for boys and girls separately. Preliminary analyses showed no evidence of a curvilinear association between negative life events and depressive symptoms.

Analyses including the protective factors Goal Orientation, Self-Confidence, Social Competence, Social Support and Family Cohesion in addition to number of negative life events and gender as predictors of depression were specified with separate regression models for each protective factor. The scores on each of the protective factors were standardized into z-scores to ease the interpretation of the results and the comparison between them. In Model 2, two-way interaction terms between the number of negative life events and each of the protective factors and between gender and the protective factors were included. In Model 3, three-way interactions of the number of negative life events, gender, and each of the protective factors were included. Age and maternal education were not included as covariates as they did not contribute significantly to the model. The analyses were adjusted for multiple comparison by using the Benjamini and Hochberg false discovery rate control [43, 44], specifying 70 comparisons and a significance level of 0.05. The p-values of < .001 and .012 remained significant (the p-value of .012 was compared to the adjusted cut-off at 0.04).

Finally, all five protective factors were included in the same regression analysis as predictors of symptoms of depression, in addition to number of negative life events. The analysis was

stratified by gender. The analyses were adjusted for multiple comparison as described above, with 12 comparisons and a significance level of 0.05. The p-values of < .001, .002 (adjusted cut-off: 0.03) and .016 (adjusted cut-off: 0.04) remained significant in the adjusted analyses. All regression models were validated using k-fold cross validation with 10 folds. The mean $R^2$ from the 10 folds is reported.

Stata 15 SE was used for the analyses [45], the Stata module CROSSFOLD was used for the k-fold cross validation [46]. Figures visualizing the results of significant interactions identified in the regression analyses were prepared in R version 3.5.1.

## Results

### Sample characteristics

There was a greater proportion of girls reporting maternal college or university education (p < .001, see Table 1). More girls reported the occurrence of negative life events than did boys (44.0% compared to 32.0%, respectively) and girls were more likely to report an increasing number of negative life events (p < .001). Regarding the specific events, there were no significant gender differences in death of a parent/guardian, sibling, close friend or girlfriend/boyfriend, while girls reported a significantly higher occurrence of the remaining negative life events (data not shown). Girls also reported a higher mean score of depressive symptoms compared to boys (7.4 compared to 4.1, respectively, p < .001). Boys reported significantly higher mean scores on all the resilience factors except social support, where girls reported higher scores (p < .001 for all resilience factors).

**Table 1. Demographic characteristics and background variables.**

| | Girls | | Boys | | | |
| | N = 5369 | | N = 4808 | | | |
| | 52.8% | | 47.2% | | | |
| | N | % | N | % | P-value | % missing |
|---|---|---|---|---|---|---|
| Age (mean (SD)) | 17.4 | 0.8 | 17.4 | 0.8 | .042 | 0.4 |
| Maternal education | | | | | < .001 | 0.7 |
| Primary school | 449 | 8.5 | 334 | 7.1 | | |
| Secondary school | 1656 | 31.3 | 1471 | 31.3 | | |
| College/university | 2015 | 38.0 | 1661 | 35.4 | | |
| Don't know | 1178 | 22.2 | 1260 | 26.2 | | |
| Negative life events | | | | | < .001 | 0.0 |
| 0 | 3005 | 56.0 | 3270 | 68.0 | | |
| 1 | 1388 | 25.9 | 1059 | 22.0 | | |
| 2 | 647 | 12.1 | 333 | 6.9 | | |
| 3 | 229 | 4.3 | 111 | 2.3 | | |
| 4 or more | 100 | 1.9 | 35 | 0.7 | | |
| Depressive symptoms (mean (SD)) | 7.4 | 6.1 | 4.1 | 4.9 | < .001 | 0.6 |
| Resilience factors (mean (SD)) | | | | | | |
| Goal Orientation | 3.75 | 0.8 | 3.86 | 0.8 | < .001 | 1.1 |
| Self-Confidence | 3.49 | 0.9 | 3.89 | 0.9 | < .001 | 1.6 |
| Social Competence | 3.84 | 0.8 | 3.92 | 0.9 | < .001 | 1.6 |
| Social Support | 4.41 | 0.7 | 4.29 | 0.8 | < .001 | 1.6 |
| Family Cohesion | 3.83 | 0.9 | 3.90 | 0.8 | < .001 | 1.9 |

SD: standard deviation. P-values are derived from chi-square tests and independent samples t-tests

## The association between negative life events and depressive symptoms

There was a positive association between number of negative life events and symptoms of depression, where an increase of one negative life event predicted a 1.86 point increase in adolescents' depression scores (p < .001) (see Table 2). The association remained significant when controlling for age, gender and maternal education (B = 1.64, p < .001).

There was a small, but significant interaction between the number of negative life events and gender, indicating that an increasing number of negative life events predicted a larger increase in depressive symptoms for girls compared to boys (see Fig 1).

## The influence of protective factors

Goal Orientation, Self-Confidence, Social Competence, Social Support, and Family Cohesion were all negatively associated with depressive symptoms when studied independently (see Table 3). Higher Self-Confidence was associated with the largest decrease in depressive symptoms, where an increase of one standard deviation (SD) in Self-Confidence was related to a decrease in depressive symptoms of -2.32 (p < .001).

In Model 2, there were significant negative interactions between the number of negative life events and Goal Orientation (Fig 2A) and Self-Confidence (Fig 2B). The interactions indicated a smaller predicted increase in depressive symptoms for an increasing number of negative life events among adolescents with higher scores on these protective factors.

Further, there were significant interactions between all the protective factors and gender, where being a girl was associated with a further decrease in depressive symptoms (Fig 3). For instance, a one SD increase in Social Support was associated with a decrease in depressive symptoms of -2.46 for girls and -1.14 for boys. The largest interaction effect was found for Social Support (β = -0.16, p< 001), while the interaction effects for the remaining protective factors were of similar size (βs of about -0.10, all p's < .001). The main effects of the protective factors were attenuated, but still significant, when the interaction terms were included.

The potential gender differences in the associations between negative life events and protective factors were further investigated by including a three-way interaction. There were no significant interactions for any of the protective factors (see Table 3).

## The relative contribution of different protective factors

Evaluating the protective factors simultaneously, three of the five factors remained as significant predictors of depressive symptoms for both genders. Specifically, Self-Confidence showed the largest negative association with depressive symptoms for both genders (β = -.29, p< 001, see Table 4). Family Cohesion was also significantly related to decreased depressive symptoms

**Table 2. Regression of number of negative life events as a predictor of symptoms of depression.**

| Independent variables | Model 1 | | | | Model 2 | | | | Model 3 | | | |
|---|---|---|---|---|---|---|---|---|---|---|---|---|
| | B | SE B | β | p-value | B | SE B | β | p-value | B | SE B | β | p-value |
| Number of NLE | 1.86 | .06 | .29 | < .001 | 1.64 | .06 | .26 | < .001 | 1.41 | .10 | .22 | < .001 |
| Age | | | | | -.03 | .06 | -.01 | .596 | -.03 | .06 | -.00 | .602 |
| Gender | | | | | 2.90 | .11 | .25 | < .001 | 2.68 | .13 | .23 | < .001 |
| NLE x gender | | | | | | | | | .38 | .13 | -.05 | .003 |
| Constant | 4.73 | .07 | | < .001 | 3.88 | 1.10 | | < .001 | 3.99 | 1.10 | | < .001 |
| Mean 10-fold $R^2$ | .09 | | | | .15 | | | | .15 | | | |

NLE: negative life events, SE: standard error

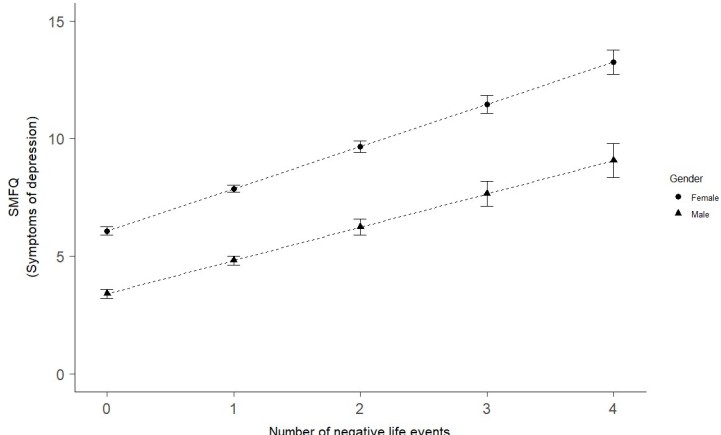

**Fig 1. The association between negative life events and symptoms of depression for boys and girls.** SMFQ: short moods and feelings questionnaire. Error bars represent 95% confidence intervals.

for both genders ($\beta$ = -.11, p< 001 for girls and $\beta$ = 0.14, p< 001 for boys). For girls, Social Support was the second most influential factor in decreasing depressive symptoms, while Family Cohesion was the second most influential factor for boys. While higher scores on Social Support were related to decreased depressive symptoms among girls ($\beta$ = -.13, p < .001), it was related to a small increase in depressive symptoms among boys ($\beta$ = .11, p = .002). Goal-Orientation was no longer significantly associated with depressive symptoms for either gender, while Social Competence showed a small negative association for girls only ($\beta$ = -.04, p < .001). The overall model explained a greater proportion of the variance in depressive symptoms among girls than boys (adjusted $R^2$ = 0.31 and 0.19, respectively).

## Discussion

Depression is a public health concern, and there is a well-established increase in depressive symptoms in adolescence [4–6]. Thus, it is important to identify factors that might protect against the development of depressive symptoms in this age group as they can form the basis of preventive interventions and help reduce the prevalence and negative consequences of depression.

In this study, we found that experiencing a higher number of negative life events was related to increases in depressive symptoms among Norwegian adolescents. Although girls reported both more symptoms of depression and more negative life events, the association between negative life events and symptoms of depression was similar for boys and girls. Findings also suggested that the impact of negative life events was somewhat larger for girls experiencing a higher number of negative events. All the potential protective factors (i.e., goal orientation, self-confidence, social competence, social support, and family cohesion) individually were associated with fewer depressive symptoms. However, interactions between number of negative life events and goal orientation and self-confidence revealed that higher scores on these protective factors were associated with a greater reduction in depressive symptoms among adolescents reporting higher numbers of negative life events. Further, significant interactions between all the protective factors and gender indicated that they were associated with a greater reduction in depression for girls than boys. When considering the protective factors jointly, self-confidence emerged as the most influential on depressive symptoms for both genders. For boys, family cohesion was the only other factor with a significant negative association with depressive symptoms; in contrast, both social support and family cohesion were of similar importance for girls.

**Table 3. Regression of negative life events and protective factors (in separate analyses) as predictors of symptoms of depression.**

| Independent variables | Model 1 | | | | Model 2 | | | | Model 3 | | | |
|---|---|---|---|---|---|---|---|---|---|---|---|---|
| | B | SE B | β | p-value | B | SE B | β | p-value | B | SE B | β | p-value |
| *Goal Orientation* | | | | | | | | | | | | |
| Number of NLE | 1.54 | .06 | .24 | < .001 | 1.51 | .06 | .24 | < .001 | 1.36 | .10 | .21 | < .001 |
| Goal Orientation | -1.62 | .05 | -.28 | < .001 | -1.11 | .08 | -.19 | < .001 | -1.10 | .09 | -.19 | < .001 |
| Gender | 2.66 | .11 | .23 | < .001 | 2.67 | .11 | .23 | < .001 | 2.53 | .13 | .22 | < .001 |
| Goal Orientation x NLE | | | | | -0.19 | .06 | -.04 | .001 | -0.22 | .09 | -.04 | .012 |
| Goal Orientation x Gender | | | | | -.73 | .11 | -.09 | < .001 | -0.77 | .13 | -.10 | < .001 |
| Goal Orientation x NLE x Gender | | | | | | | | | 0.07 | .11 | .01 | .528 |
| Constant | 3.50 | .08 | | < .001 | 3.47 | .09 | | < .001 | 3.55 | .09 | | < .001 |
| Mean 10-fold $R^2$ | 0.22 | | | | 0.23 | | | | 0.23 | | | |
| *Self-Confidence* | | | | | | | | | | | | |
| Number of NLE | 1.44 | .06 | .23 | < .001 | 1.37 | .06 | .22 | < .001 | 1.32 | .09 | .21 | < .001 |
| Self-Confidence | -2.32 | .05 | -.40 | < .001 | -1.65 | .08 | -.29 | < .001 | -1.63 | .09 | -.28 | < .001 |
| Gender | 1.86 | .10 | .16 | < .001 | 1.91 | .10 | .16 | < .001 | 1.85 | .12 | .16 | < .001 |
| Self-Confidence x NLE | | | | | -0.31 | .05 | -.06 | < .001 | -0.36 | .09 | -.07 | < .001 |
| Self-Confidence x Gender | | | | | -0.80 | .10 | -.10 | < .001 | -0.85 | .13 | -.11 | < .001 |
| Self-Confidence x NLE x Gender | | | | | | | | | 0.09 | .11 | .02 | .408 |
| Constant | 4.00 | .08 | | < .001 | 3.89 | .08 | | < .001 | 3.92 | .09 | | < .001 |
| Mean 10-fold $R^2$ | 0.30 | | | | 0.30 | | | | 0.29 | | | |
| *Social Competence* | | | | | | | | | | | | |
| Number of NLE | 1.69 | .06 | .27 | < .001 | 1.69 | .06 | .26 | < .001 | 1.47 | .10 | .23 | < .001 |
| Social Competence | -1.75 | .05 | -.30 | < .001 | -1.34 | .08 | -.23 | < .001 | -1.31 | .09 | -.23 | < .001 |
| Gender | 2.70 | .11 | .23 | < .001 | 2.70 | .11 | .23 | < .001 | 2.49 | .13 | .21 | < .001 |
| Social Competence x NLE | | | | | -0.04 | .06 | -.01 | .483 | -0.08 | .09 | -.02 | .362 |
| Social Competence x Gender | | | | | -0.75 | .11 | -.09 | < .001 | -0.81 | .13 | -.10 | < .001 |
| Social Competence x NLE x Gender | | | | | | | | | 0.08 | .12 | .01 | .478 |
| Constant | 3.38 | .08 | | < .001 | 3.37 | .08 | | < .001 | 3.48 | .09 | | < .001 |
| Mean 10-fold $R^2$ | 0.24 | | | | 0.24 | | | | 0.24 | | | |
| *Social Support* | | | | | | | | | | | | |
| Number of NLE | 1.49 | .06 | .23 | < .001 | 1.44 | .06 | .23 | < .001 | 1.36 | .10 | .21 | < .001 |
| Social Support | -1.76 | .05 | -.30 | < .001 | -1.14 | .08 | -.20 | < .001 | -1.12 | .09 | -.19 | < .001 |
| Gender | 3.18 | .11 | .27 | < .001 | 3.19 | .11 | .27 | < .001 | 3.11 | .13 | .27 | < .001 |
| Social Support x NLE | | | | | 0.00 | .05 | .00 | .934 | -0.03 | .08 | -.01 | .756 |
| Social Support x Gender | | | | | -1.32 | .11 | -.16 | < .001 | -1.35 | .13 | -.16 | < .001 |
| Social Support x NLE x Gender | | | | | | | | | 0.05 | .11 | .01 | .644 |
| Constant | 3.26 | .08 | | < .001 | 3.34 | .08 | | < .001 | 3.38 | .09 | | < .001 |
| Adjusted $R^2$ | 0.24 | | | | 0.25 | | | | 0.25 | | | |
| Mean 10-fold $R^2$ | 0.24 | | | | 0.25 | | | | 0.25 | | | |
| *Family Cohesion* | | | | | | | | | | | | |
| Number of NLE | 1.24 | .06 | .29 | < .001 | 1.22 | .06 | .19 | < .001 | 1.16 | .10 | .28 | < .001 |
| Family Cohesion | -1.96 | .05 | -.34 | < .001 | -1.52 | .08 | -.26 | < .001 | -1.44 | .09 | -.25 | < .001 |
| Gender | 2.77 | .10 | .24 | < .001 | 2.78 | .10 | .24 | < .001 | 2.75 | .13 | .24 | < .001 |
| Family Cohesion x NLE | | | | | 0.02 | .05 | .01 | .636 | -0.10 | .09 | -.02 | .228 |
| Family Cohesion x Gender | | | | | -0.83 | .11 | -.11 | < .001 | -0.97 | .13 | -.12 | < .001 |
| Family Cohesion x NLE x Gender | | | | | | | | | 0.20 | .11 | 0.04 | .060 |
| Constant | 3.62 | .08 | | < .001 | 3.62 | .08 | | < .001 | 3.63 | .09 | | < .001 |
| Mean 10-fold $R^2$ | 0.25 | | | | 0.26 | | | | 0.26 | | | |

NLE: negative life events, SE: standard error. Analyses were conducted separately for the individual protective factors.

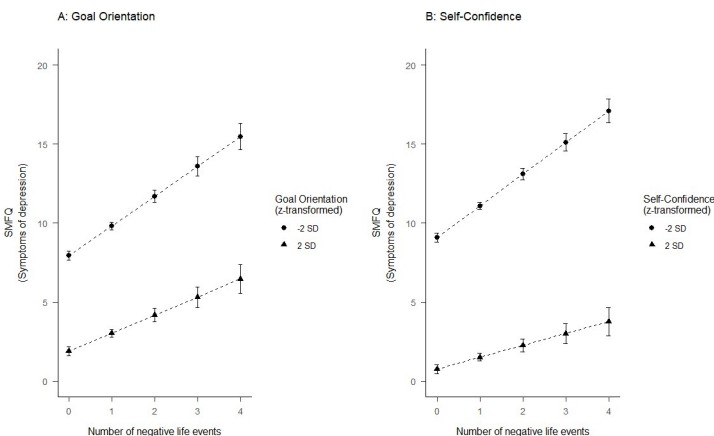

**Fig 2.** The association between negative life events and symptoms of depression for adolescents with (A) Goal Orientation and (B) Self-Confidence scores ±2 standard deviations from the mean score. SMFQ: short moods and feelings questionnaire, SD: standard deviation. Error bars represent 95% confidence intervals.

The association between negative life events and depressive symptoms is in line with previous research [9–11]. Similar to previous studies, we found that girls reported more symptoms of depression [5, 8] and a higher number of negative life events [29] compared to boys. Still, the association between negative life events and depressive symptoms was similar for boys and girls, and appeared to be mostly linear. This finding differs from a previous study where a threshold of increased risk for depression appeared at three negative life events, with little difference in depressive symptoms between adolescents reporting 0 to 2 negative life events and between adolescents reporting 3 or more negative events [11]. This difference across studies could be due to the different measures used; for example, the previous study investigated risk for a diagnosable disorder and all events occurred within a one-year period. The mostly linear relationship identified in the present study is consistent with a study of the impact of negative life events and family stress on mental health problems in younger children [47].

When studied separately, all the potentially protective factors were negatively associated with symptoms of depression. This finding is consistent with previous studies using the

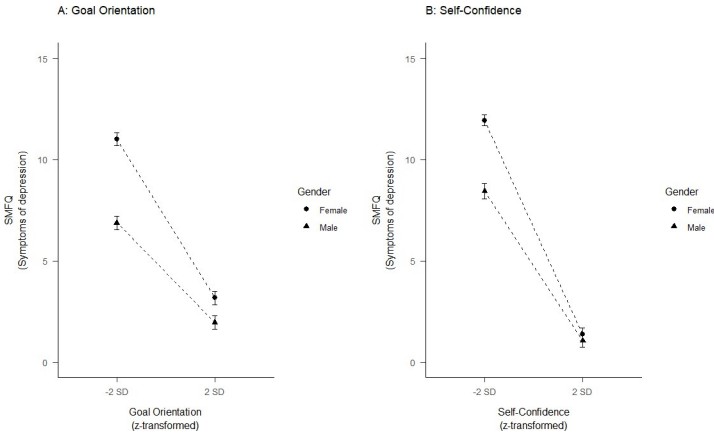

**Fig 3.** The association between symptoms of depression and (A) Goal Orientation and (B) Self-Confidence for girls and boys. SMFQ: short moods and feelings questionnaire, SD: standard deviation. Error bars represent 95% confidence intervals.

**Table 4. Regression of negative life events and protective factors as predictors of symptoms of depression.**

| Independent variables | Girls | | | | Boys | | | |
|---|---|---|---|---|---|---|---|---|
| | B | SE B | β | p-value | B | SE B | β | p-value |
| Number of NLE | 1.35 | .08 | .22 | < .001 | 1.22 | .09 | .20 | < .001 |
| Goal Orientation | 0.05 | .10 | .01 | .588 | 0.08 | .10 | .02 | .429 |
| Self-Confidence | -1.77 | .11 | -.29 | < .001 | -1.49 | .12 | -.29 | < .001 |
| Social Competence | -0.24 | .10 | -.04 | .016 | -0.16 | .11 | -.03 | .122 |
| Social Support | -0.84 | .11 | -.13 | < .001 | 0.32 | .11 | .07 | .002 |
| Family Cohesion | -0.67 | .11 | -.11 | < .001 | -0.71 | .12 | -.14 | < .001 |
| Constant | 6.06 | .09 | | < .001 | 3.97 | .09 | | < .001 |
| Mean 10-fold $R^2$ | 0.31 | | | | 0.19 | | | |

NLE: negative life events, SE: standard error.

original READ factors [21, 27, 48, 49] and studies of similar personal factors [19, 23] and measures of family cohesion and social support [18–20].

The protective model of resilience (i.e., that protective factors are more influential in the face of risk) was supported for goal orientation and self-confidence, indicating that these factors were especially protective for adolescents who experienced a higher number of negative life events. The effects were small, which is a common finding in resilience research (i.e., the addition of an interaction term does not explain much variance), possibly because there is little variance left to be explained after accounting for main effects [37]. Still, as the sample size in the present study is large, we were able to detect the interaction effects with the personal protective factors. As the process of resilience varies according to the specific risks and outcomes being studied [16], it is possible that investigating measures of more chronic stress, instead of focusing solely on discrete negative events, might yield even stronger support for protective models of resilience [22]. It is also possible that including other negative life events, such as parental divorce or academic difficulties could yield different results. Further, longitudinal studies are needed to gain a more complete understanding of the influence of negative life events on depressive symptoms and the possible protective influence of resilience factors across adolescent development.

The present study mainly supports a compensatory model of resilience. Adolescents with greater goal orientation, self-confidence, social competence, social support and family cohesion reported lower symptoms of depression, regardless of their exposure to negative life events. Similar results have been found for the original READ factors [21], for a study of self-competence [10], and in a meta-analysis focusing on social support [22]. Together, the data suggest that interventions aimed at boosting these potential protective factors would be beneficial for all adolescents, not only for those facing adverse circumstances. Given the high prevalence of depressive symptoms in adolescence, the present findings suggest that preventive interventions delivered at the universal level could be beneficial. There is some evidence that universal interventions can be effective in preventing depressive symptoms [50].

Interestingly, the compensatory effect of the protective factors was especially strong for girls; all the protective factors were associated with lower levels of depressive symptoms among girls. In the context of depressive symptoms and depression, girls could be viewed as being an "at risk" population. Previous studies have found that preventive interventions often work better for adolescents at higher risk for depressive symptoms [50], and the present findings indicate that interventions aimed at boosting protective factors related to resilience could be especially effective among girls. It is important to emphasize, however, that this finding is specific to the investigation of depressive symptoms, as the importance of protective factors is expected to vary

according to the outcome being studied [16]. Thus, one should not conclude that protective factors related to resilience are more important for girls than for boys in general.

Investigating the relative contributions of the protective factors revealed that self-confidence emerged clearly as the most influential factor for both boys and girls. Thus, self-confidence works both as a buffer and a main effect for all adolescents, indicating that interventions designed to increase self-confidence among adolescents could help them deal with adversity. This finding is in line with a study demonstrating the importance of personal protective factors in explaining youths' depressive symptoms [25]. It could be argued that the importance of self-confidence in explaining depressive symptoms could be because these two concepts are contrary to each other, that is, feeling competent and having a positive outlook despite hardship are quite opposite to depressive symptoms such as lack of energy, pessimism, and helplessness. Still, although self-confidence is negatively correlated with emotional problems, the correlation is moderate [30]. Further, previous research on the original READ factor of personal competence (which includes items measuring self-confidence) indicate that it can predict both depressive and social anxiety symptoms, and is not simply measuring the opposite of depression [21].

Our findings significantly extend prior research by demonstrating that family cohesion remained a significant protective factor for boys and girls, even when considering the influence of personal protective factors. This finding is consistent with literature showing that family influences remain strong throughout adolescence [22, 51] and that family support is equally important in explaining depressive symptoms for boys and girls [52]. It is further in line with the resilience literature, where a close relationship with parents, or other competent adults, is described as the most important protective factor for youth development [17, 53]. Together, the findings from the present study suggest that efforts to build self-confidence and also promote family cohesion might be good strategies for fostering resilience in adolescents–both for boys and girls. There is evidence that preventive interventions can lead to changes in family functioning with long-term benefits in preventing adolescent depression, and it is recommended that efforts to strengthen family functioning should be included in such interventions [50].

In contrast, social support was related to fewer depressive symptoms among girls and greater symptoms among boys. For girls, the results are in line with a previous study [34]. However, the finding for boys is more puzzling, and in contrast to a meta-analysis where no gender differences were detected for the importance of social support in explaining depression [22]. It is possible that the inclusion of personal protective factors in the present study mitigated the additional (typically favorable) impact of social support for boys. This interpretation is in line with a study where social support did not discriminate significantly between groups when self-efficacy and coping strategies were taken into account [23]. Similarly, the inclusion of a key family factor–family cohesion–in the present study, might have mitigated the impact of social support for boys. Family cohesion has consistently been shown to be of importance in preventing depression in adolescence, while the findings regarding social support from peers are varied [22]. This underlines the importance of including several resilience factors in the same analysis to gain further knowledge of their independent contributions. Future research is needed to evaluate the contributions of social support as a protective factor for boys' depressive symptoms.

## Strengths and limitations

Strengths of the study include a population-based design with a large sample size and the inclusion of validated measures of symptoms of depression and protective factors associated with resilience. Further, the large sample size ensures sufficient power to investigate interactions. Another strength is the investigation of the relative contributions of potential protective factors in predicting symptoms of depression.

A limitation of the present study is the cross-sectional nature of the data. Although it is likely that the majority of negative life events took place before the self-reports of depression and protective factors (depression and protective factors were reported for the past two weeks and the past month, respectively), they were all reported at the same time. Thus, the temporal associations between these variables are uncertain. Still, the proposed direction is supported by previous studies detecting associations between negative life events and depression also when controlling for earlier levels of depressive symptoms [54, 55], while there was no significant association between depressive symptoms and future life events [55]. The directionality could depend on the specific stressors investigated. For example, it is possible that depressive symptoms are more important in predicting minor negative events, such as peer problems, but not major stressful events, such as the death of loved ones. With regards to the associations between protective factors and depression, there are longitudinal studies indicating that family support predicted later depression and depressive symptoms, not the opposite direction [35, 52]. Another study found that family cohesion, a positive self-concept and more positive relations with others measured in adolescence predicted depression in early adulthood [19]. Still, it is possible that some protective factors could be influenced by depression, as depressive symptoms have been found to predict less peer support in adolescence [35]. Due to the cross-sectional nature of the data, it is impossible to ascertain whether the READ factors did indeed serve as protective factors. They should therefore be viewed as potential protective factors based on the present findings.

A further limitation is the low participation rate of 53%, which could lead to sampling bias. It is possible that adolescents with mental health problems and/or adolescents who had experienced several negative life events were less likely to participate in the survey. Thus, the prevalence estimates for depressive symptoms and negative life events could be underestimated in the current study, though it has been suggested that measures of association are less affected by selective participation [56]. As the present study included adolescents from the general population, where only a small percentage had experienced several negative life events, it is a pertinent question whether the results would be different in a more vulnerable population. Further, only late adolescents were included in the study and the findings cannot be generalized to younger adolescents. It is further possible that positive life events may have a buffering effect on the association between negative life events and emotional distress [57], and it is possible that inclusion of positive life events could lead to more nuanced findings.

## Conclusion

The present study supports a compensatory model of resilience, where the potential protective factors goal orientation, self-confidence, social competence, social support and family cohesion were all related to a decrease in depressive symptoms, with similar effects for different levels of negative life events. Further, the results suggest self-confidence and family cohesion as especially important protective factors to target when aiming to reduce symptoms of depression in adolescence. Interestingly, all the protective factors were associated with larger decreases in depressive symptoms among girls than boys. Thus, while girls report more symptoms of depression, they might also be more likely to benefit from an increase in the protective factors investigated in the present study.

## Supporting information

**S1 Table. The resilience scale for adolescents (READ).** Abbreviated wording of items adapted from Von Soest et al., 2010.
(DOCX)

## Acknowledgments

The authors are grateful to all participants who made this study possible and would also like to thank the Bergen Child Study group.

## Author Contributions

**Conceptualization:** Kristin Gärtner Askeland, Kyrre Breivik, Annette M. La Greca, Børge Sivertsen, Mari Hysing.

**Data curation:** Kristin Gärtner Askeland, Tormod Bøe, Mari Hysing.

**Formal analysis:** Kristin Gärtner Askeland, Tormod Bøe, Kyrre Breivik.

**Funding acquisition:** Kristin Gärtner Askeland, Børge Sivertsen, Mari Hysing.

**Investigation:** Tormod Bøe, Mari Hysing.

**Supervision:** Mari Hysing.

**Visualization:** Tormod Bøe.

**Writing – original draft:** Kristin Gärtner Askeland.

**Writing – review & editing:** Kristin Gärtner Askeland, Tormod Bøe, Kyrre Breivik, Annette M. La Greca, Børge Sivertsen, Mari Hysing.

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
