## [Decision Letter · Decision Letter 0]

21 Feb 2020

PONE-D-19-26146

Life events and adolescent depressive symptoms: Protective factors associated with resilience

PLOS ONE

Dear Dr Askeland,

Thank you for submitting your manuscript to PLOS ONE. After careful consideration, we feel that it has merit but does not fully meet PLOS ONE’s publication criteria as it currently stands. Therefore, we invite you to submit a revised version of the manuscript that addresses the points raised during the review process.

The two reviewers addressed a number of major and minor concerns about your manuscript. Please revise your manuscript carefully.

We would appreciate receiving your revised manuscript by Apr 06 2020 11:59PM. To enhance the reproducibility of your results, we recommend that if applicable you deposit your laboratory protocols in protocols.io, where a protocol can be assigned its own identifier (DOI) such that it can be cited independently in the future. For instructions see: http://journals.plos.org/plosone/s/submission-guidelines#loc-laboratory-protocols

We look forward to receiving your revised manuscript.

Kind regards,

Kenji Hashimoto, PhD

Academic Editor

PLOS ONE

Journal Requirements:

2. We note that you have stated that "In accordance Norwegian regulations, adolescents aged 16 years and older can make decisions regarding their own health (including participation in health studies), and thus gave consent themselves to participate in the current study." Please clarify whether your REC approved the lack of consent from parents/guardians in your study. Please include this information in the Methods section.

Reviewers' comments:

Reviewer's Responses to Questions

**Comments to the Author**

1. Is the manuscript technically sound, and do the data support the conclusions?

Reviewer #1: Yes

Reviewer #2: Partly

2. Has the statistical analysis been performed appropriately and rigorously? 

Reviewer #1: Yes

Reviewer #2: No

3. Have the authors made all data underlying the findings in their manuscript fully available?

Reviewer #1: No

Reviewer #2: Yes

4. Is the manuscript presented in an intelligible fashion and written in standard English?

Reviewer #1: Yes

Reviewer #2: Yes

5. Review Comments to the Author

Reviewer #1: The authors investigated the association between negative life events and depressive symptoms among adolescents, and examined the influence and relative contributions of personal, social and family related protective factors. They found that experiencing a higher number of negative life events was related to increases in depressive symptoms, while the protective factors goal orientation, self-confidence, social competence, social support, and family cohesion individually were associated with fewer symptoms. When considering the protective factors jointly, only self-confidence emerged as the most influential predictor of depressive symptoms for both genders. These findings will be of interest to clinicians, as well as researchers in the field.

I have the following concerns.

#1. Introduction, P5, line 94. “Another important consideration is the possibility of gender difference.”

I think it would be useful if the authors gave more information about the gender difference in the protective factors related to resilience. Are there well-known gender differences in the protective factors related to resilience?

#2. Methods. P10, line 210. “Gender differences in age and symptoms of depression were investigated by independent samples t-test,”

Theoretically, a t-test is applied when the variables would follow a normal distribution. However, according to your previous study, the distributions of depressive symptom scores (SMFQ) are right-skewed and may follow an exponential distribution, except for the lower end of the distributions (Lundervold AJ,et al. 2013 Front. Psychol. 4:613). Additionally, recent research reports that total scores on these depressive symptom scales follow an exponential distribution, except for the lower end of the distribution (Distributional patterns of item responses and total scores on the PHQ-9 in the general population: data from the National Health and Nutrition Examination Survey- BMC psychiatry, 2018).

Even if variables exhibit a non-normal distribution, researchers often use statistical methods that assume a normal distribution. Thus, I think that there is no need to use other statistical methods, such as non-parametric statistics. But I think it would be useful if the authors gave some information on the right-skewed pattern of the SMFQ scores and the robustness of t-test.

#4. Results Table 1

Table1 shows the gender difference both in depressive symptoms and in the number of negative events. I think it would be useful if the authors gave some values about the gender difference in the protective factors related to resilience.

#3. Discussion, P24, line 409. “It could be argued that the importance of self-confidence in explaining depressive symptoms could be due to the similarities between these two concepts.”

“the similarities between these two concepts” just doesn't feel right to me. I think it is rather “opposite concepts” than “similar concepts”.

In conclusion, I enjoyed reading this paper. This is a valuable paper that presents the relationship between negative life events, protective factors and depressive symptoms among adolescents. In addition, there are significant interactions between the protective factors and gender Interestingly, while girls report more symptoms of depression, they might also be more likely to benefit from an increase in the protective factors.

Finally, I am grateful that the authors and editor have given me this kind of opportunity. I think these findings help prevent development of depressive symptoms in adolescence.

Reviewer #2: This study examined data from a cross-sectional survey of adolescents with respect to depressive symptoms, negative life events and factors associated with resilience. The authors report a significant association between greater number of negative life events and increased depressive symptoms as well as ‘protective’ effects of factors implicated in resilience, with a significant interaction with gender. Additionally, the authors report on findings from regression modeling. A major strength of the study is the relatively large sample size, and major weaknesses include (1) the cross-sectional nature of the survey data acquired, (2) lack of test/train or cross-validation approach to the regression models, and (3) unclear whether correction for multiple regression models/tests was conducted which limit the claims that can be made with respect to study findings. Nonetheless, this is an understudied and important area of research that can help define and develop strategies to augment resilience and prevent depression during a vulnerable developmental period. Nevertheless the results should be interpreted more cautiously.

The following are points for clarification and/or additional information:

Abstract and Introduction

The authors state that a primary aim is to identify “protective factors” for adolescent depression. Given that the data is cross sectional, the word protective may be misleading, and changing to factors implicated in resilience seems more appropriate.

Similarly, the authors note a focus on defining significant “predictors” of depressive symptoms in youth but given the cross-sectional data. Make sentence in abstract clearer — (lines 39-41) — direction of association between self-confidence and family cohesion and depressive symptoms

Methods

Other than age gender and maternal education, were other demographics collected? If so, please include in the demographics section.

More clarity on risk factors such as parental diagnosis (e.g. having a depressed parent) would be informative in testing the proposed associations and models, in addition to the experience of negative life events.

It does not appear that the authors corrected for multiple comparisons and models and ran multiple correlations and predictive models. The authors should correct for the analyses conducted.

Regression models should be tested using a test/training split sample or k-fold cross validation rather than tested in the whole sample as a method of validation.

Results

The authors should report on specific life events that significantly differed by gender and accounted for the p<0.001 difference— e.g. was this a significant difference in death of a loved one or sexual abuse which affect interpretation of study findings.

In the regression tables, the authors list ‘predictor’ variables, while the may be technically correct a less confusing terminology would be ‘independent variables’ given that this data was acquired cross sectionally.

Discussion

Please add references for “increase in depressive symptoms in adolescence” (lines 330-331).

In the introduction and in the discussion the authors should tone down the claims with respect to ‘protective factors’ given the cross-sectional nature of the associations found between factors implicated in resilience and depressive symptoms.

Please clarify what is meant by “study where a sharp increase in depression appeared at three negative life events” (line 357).

Paragraph beginning on line 353: please update and provide amore comprehensive review of the literature on the relationship between negative life events (e.g. abuse, death of parent) and depression; currently, only one study from 1999 is referenced.

It would be informative to broaden the scope of the discussion in paragraph 5 beginning on line 376 to not only discuss chronic stress but other specific negative life events not examined in the present study as well as the longitudinal assessment of risk and influence of negative life events across adolescent development.

The authors should consider changing “protective” factors to potential protective factors or resilience factors given that these measures were collected cross-sectionally at the same time as when depressive symptoms were assessed.

Additional limitations should be noted including (1) the possibility of a sampling bias, (2) adolescents surveyed in the analysis were late in adolescence, and the study did not examine younger adolescents where there may be different relations between resilience and risk factors and depressive symptoms, (3) the influence of positive life events was not assessed and could have a moderating effect on depression or resilience (e.g. Fischer et al JAMA Psychiatry 2018).

Conclusion - please remove the word “predicted” from lines 478 and 482, prediction is not possible with this cross-sectional data set.

Minor Points

Unclear meaning of the sentence beginning on line 66 “positive development is in itself not sufficient to establish that resilience is present.” Please explain and provide references.

In might be helpful to provide further detail on gender differences in the experience of negative life events (lines 95-96).

Please insert stats for Family Cohesion (beta, p) in lines 313-314 to be consistent with remainder of results reported.

6. PLOS authors have the option to publish the peer review history of their article (what does this mean?). If published, this will include your full peer review and any attached files.

Reviewer #1: No

Reviewer #2: No

---

## [Author Response · Author response to Decision Letter 0]

6 May 2020

Dear editor,

Thank you for the opportunity to revise and resubmit the manuscript ‘Life events and adolescent depressive symptoms: Protective factors associated with resilience’. We appreciate the reviewers’ thorough and valuable comments. We have revised the manuscript following their suggestions and below you will find our responses to the comments from the reviewers. 

We look forward to hearing from you in due course.

Reviewers’ comments:

Reviewer #1: The authors investigated the association between negative life events and depressive symptoms among adolescents, and examined the influence and relative contributions of personal, social and family related protective factors. They found that experiencing a higher number of negative life events was related to increases in depressive symptoms, while the protective factors goal orientation, self-confidence, social competence, social support, and family cohesion individually were associated with fewer symptoms. When considering the protective factors jointly, only self-confidence emerged as the most influential predictor of depressive symptoms for both genders. These findings will be of interest to clinicians, as well as researchers in the field.

I have the following concerns.

#1. Introduction, P5, line 94. “Another important consideration is the possibility of gender difference.”

I think it would be useful if the authors gave more information about the gender difference in the protective factors related to resilience. Are there well-known gender differences in the protective factors related to resilience?

Response:

Thank you for pointing this out. Some gender differences have been identified, and have been included in the introduction:

‘Regarding protective factors related to resilience, studies suggest that girls have higher scores on social competence (27, 30-32), while boys have higher scores on self-confidence (27, 30-32) and self-esteem (33). It is therefore important to investigate whether such gender differences also exist in the influence of potential protective factors on the association between negative life events and symptoms of depression.’

#2. Methods. P10, line 210. “Gender differences in age and symptoms of depression were investigated by independent samples t-test,”

Theoretically, a t-test is applied when the variables would follow a normal distribution. However, according to your previous study, the distributions of depressive symptom scores (SMFQ) are right-skewed and may follow an exponential distribution, except for the lower end of the distributions (Lundervold AJ,et al. 2013 Front. Psychol. 4:613). Additionally, recent research reports that total scores on these depressive symptom scales follow an exponential distribution, except for the lower end of the distribution (Distributional patterns of item responses and total scores on the PHQ-9 in the general population: data from the National Health and Nutrition Examination Survey- BMC psychiatry, 2018).

Even if variables exhibit a non-normal distribution, researchers often use statistical methods that assume a normal distribution. Thus, I think that there is no need to use other statistical methods, such as non-parametric statistics. But I think it would be useful if the authors gave some information on the right-skewed pattern of the SMFQ scores and the robustness of t-test.

Response:

Thank you for pointing this out. We agree with these considerations, and have added the following to the statistical methods:

‘The SMFQ scores had a right skewed distribution deviating from normality (skewness 1.3, kurtosis 4.3). As the independent samples t-test is robust to deviations from the assumption of a normal distribution in large samples (42), gender differences in symptoms of depression were nevertheless investigated by independent samples t-test.’

#4. Results Table 1

Table1 shows the gender difference both in depressive symptoms and in the number of negative events. I think it would be useful if the authors gave some values about the gender difference in the protective factors related to resilience.

Response: Thank you for the suggestion, we have now added the gender difference on the protective factors in table 1. 

#3. Discussion, P24, line 409. “It could be argued that the importance of self-confidence in explaining depressive symptoms could be due to the similarities between these two concepts.”

“the similarities between these two concepts” just doesn't feel right to me. I think it is rather “opposite concepts” than “similar concepts”.

Response:

We agree, the sentence now reads: 

‘It could be argued that the importance of self-confidence in explaining depressive symptoms could be because these two concepts are contrary to each other, where feeling competent and having a positive outlook despite hardship is quite opposite to depressive symptoms such as lack of energy, pessimism, and helplessness.’

In conclusion, I enjoyed reading this paper. This is a valuable paper that presents the relationship between negative life events, protective factors and depressive symptoms among adolescents. In addition, there are significant interactions between the protective factors and gender Interestingly, while girls report more symptoms of depression, they might also be more likely to benefit from an increase in the protective factors.

Finally, I am grateful that the authors and editor have given me this kind of opportunity. I think these findings help prevent development of depressive symptoms in adolescence.

Reviewer #2: This study examined data from a cross-sectional survey of adolescents with respect to depressive symptoms, negative life events and factors associated with resilience. The authors report a significant association between greater number of negative life events and increased depressive symptoms as well as ‘protective’ effects of factors implicated in resilience, with a significant interaction with gender. Additionally, the authors report on findings from regression modeling. A major strength of the study is the relatively large sample size, and major weaknesses include (1) the cross-sectional nature of the survey data acquired, (2) lack of test/train or cross-validation approach to the regression models, and (3) unclear whether correction for multiple regression models/tests was conducted which limit the claims that can be made with respect to study findings. Nonetheless, this is an understudied and important area of research that can help define and develop strategies to augment resilience and prevent depression during a vulnerable developmental period. Nevertheless the results should be interpreted more cautiously.

The following are points for clarification and/or additional information:

Abstract and Introduction

The authors state that a primary aim is to identify “protective factors” for adolescent depression. Given that the data is cross sectional, the word protective may be misleading, and changing to factors implicated in resilience seems more appropriate.

Response: Thank you for pointing this out, the aim now reads:

‘This study aimed to investigate the association between negative life events and depressive symptoms among adolescents, and to examine the influence and relative contributions of personal, social and family factors related to resilience.’

Similarly, the authors note a focus on defining significant “predictors” of depressive symptoms in youth but given the cross-sectional data. Make sentence in abstract clearer — (lines 39-41) — direction of association between self-confidence and family cohesion and depressive symptoms

Response: We have included the direction of the association, and the sentence now reads:

 ‘When considering the factors implicated in resilience jointly, only self-confidence and family cohesion were significantly associated with fewer depressive symptoms for both genders, with the addition of social support for girls.’

Methods

Other than age gender and maternal education, were other demographics collected? If so, please include in the demographics section.

Response: The youth@hordaland-survey is comprehensive, and the questionnaire also included other demographics, such as ethnicity, living situation and paternal education. We included age, gender and maternal education in the demographics sections as these are the demographic variables included in the dataset we applied for and received to conduct the analyses for the present study. 

More clarity on risk factors such as parental diagnosis (e.g. having a depressed parent) would be informative in testing the proposed associations and models, in addition to the experience of negative life events.

Response: We agree that this would be informative, unfortunately, information on parental diagnosis was not included in the youth@hordaland-survey. 

It does not appear that the authors corrected for multiple comparisons and models and ran multiple correlations and predictive models. The authors should correct for the analyses conducted.

Response: Thank you for pointing this out, we have corrected for multiple comparisons using the Benjamini and Hochberg false discovery rate control. The following is included in the methods section:

‘The analyses were adjusted for multiple comparison by using the Benjamini and Hochberg false discovery rate control (43, 44), specifying 70 comparisons and a significance level of 0.05. The p-values of <.001 and .012 remained significant (the p-value of .012 was compared to the adjusted cut-off at 0.04).’

‘The analyses were adjusted for multiple comparison as described above, with 12 comparisons and a significance level of 0.05. The p-values of <.001, .002 (adjusted cut-off: 0.03) and .016 (adjusted cut-off: 0.04) remained significant in the adjusted analyses.’

Regression models should be tested using a test/training split sample or k-fold cross validation rather than tested in the whole sample as a method of validation.

Response: Thank you for the suggestion, we have conducted k-fold cross validation with 10 folds for the regression analyses. The mean R2 from the 10 folds are reported in table 2, 3 and 4, and the following is included in the methods section: 

‘All regression models were validated using k-fold cross validation with 10 folds. The mean R2 from the 10 folds is reported.’

Results

The authors should report on specific life events that significantly differed by gender and accounted for the p<0.001 difference— e.g. was this a significant difference in death of a loved one or sexual abuse which affect interpretation of study findings.

Response: Thank you for the suggestion. We investigated gender differences in the specific negative life events, and included the following sentence in the results:

‘Regarding the specific events, there were no significant gender differences in death of a parent/guardian, sibling, close friend or girlfriend/boyfriend, while girls reported a significantly higher occurrence of the remaining negative life events (data not shown).’

In the regression tables, the authors list ‘predictor’ variables, while the may be technically correct a less confusing terminology would be ‘independent variables’ given that this data was acquired cross sectionally.

Response: Thank you for pointing this out, the suggested changes have been made to the regression tables.

Discussion

Please add references for “increase in depressive symptoms in adolescence” (lines 330-331).

Response: References have now been included:

‘4. Hyde JS, Mezulis AH, Abramson LY. The ABCs of depression: integrating affective, biological, and cognitive models to explain the emergence of the gender difference in depression. Psychologial Review. 2008;115(2):291-313.

5. Green H, McGinnity A, Meltzer H, Ford T, Goodman R. Mental health of children and young people in Great Britan, 2004. Basingstroke, UK: Palgrave Macmillan; 2005.

6. Salk RH, Petersen JL, Abramson LY, Hyde JS. The contemporary face of gender differences and similarities in depression throughout adolescence: Development and chronicity. Journal of affective disorders. 2016;205:28-35.’

In the introduction and in the discussion the authors should tone down the claims with respect to ‘protective factors’ given the cross-sectional nature of the associations found between factors implicated in resilience and depressive symptoms.

Response: Thank you for pointing this out. In a later comment, the reviewer suggests calling these factors potential protective factors or resilience factors. 

Resilience research and theory describe resilience as a process, not a trait, and in our opinion, it is more accurate to use the term protective factor rather than resilience factor. We have therefore chosen not to use the term resilience factor, due to the cross-sectional nature of the data. We agree that the suggested ‘potential protective factor’ acknowledges the limitations of cross-sectional data, however, we found that including this term throughout the manuscript instead of protective factors interferes with readability. We therefore suggest the following changes that have been applied to the manuscript: 

Include the suggested change of potential protective factors in the abstract, but not in the second mention of protective factors in the same sentence. 

Include a clarification in the introduction:

‘Research into potential protective factors (henceforth called protective factors) for adolescents at risk for developing depression have included all these domains, and social support has received particular attention.’

Include the limitations of cross-sectional data in the instruments section:

‘Though the READ factors are more accurately described as potential protective factors due to the cross-sectional design of the study, they will be referred to as protective factors in the following to ease readability.’

Include this consideration also in the limitations section:

‘Due to the cross-sectional nature of the data, it is impossible to ascertain whether the READ factors did indeed serve as protective factors. They should therefore be viewed as potential protective factors based on the present findings.’

In addition, we have referred to the READ factors as potential protective factors in the first mention in the discussion section and in the conclusion:

‘All the potential protective factors (i.e., goal orientation, self-confidence, social competence, social support, and family cohesion) individually were associated with fewer depressive symptoms.’

‘The present study supports a compensatory model of resilience, where the potential protective factors goal orientation, self-confidence, social competence, social support and family cohesion were all related to a decrease in depressive symptoms, with similar effects for different levels of negative life events.’

We hope that including these changes to the manuscript will help to tone down the claims, as suggested by the reviewer. 

Please clarify what is meant by “study where a sharp increase in depression appeared at three negative life events” (line 357).

Response: We have rewritten the sentence, and it now reads:

‘This finding differs from a previous study where a threshold of increased risk for depression appeared at three negative life events, with little difference in depressive symptoms between adolescents reporting 0 to 2 negative life events and between adolescents reporting 3 or more negative events (11).’

Paragraph beginning on line 353: please update and provide amore comprehensive review of the literature on the relationship between negative life events (e.g. abuse, death of parent) and depression; currently, only one study from 1999 is referenced.

Response: Thank you for pointing this out, we have included references on the relationship between negative life events and depression in general in addition to the references regarding gender differences.

‘The association between negative life events and depressive symptoms is in line with previous research (9-11). Similar to previous studies, we found that girls reported more symptoms of depression (5, 8) and a higher number of negative life events (29) compared to boys.’

It would be informative to broaden the scope of the discussion in paragraph 5 beginning on line 376 to not only discuss chronic stress but other specific negative life events not examined in the present study as well as the longitudinal assessment of risk and influence of negative life events across adolescent development.

Response: Thank you for the suggestions, we have included the following at the end of paragraph 5 in the discussion:

‘It is also possible that including other negative life events, such as parental divorce or academic difficulties could yield different results. Further, longitudinal studies are needed to gain a more complete understanding of the influence of negative life events on depressive symptoms and the possible protective influence of resilience factors across adolescent development.’

The authors should consider changing “protective” factors to potential protective factors or resilience factors given that these measures were collected cross-sectionally at the same time as when depressive symptoms were assessed.

Response: Please see response above to similar comment. 

Additional limitations should be noted including (1) the possibility of a sampling bias, (2) adolescents surveyed in the analysis were late in adolescence, and the study did not examine younger adolescents where there may be different relations between resilience and risk factors and depressive symptoms, (3) the influence of positive life events was not assessed and could have a moderating effect on depression or resilience (e.g. Fischer et al JAMA Psychiatry 2018).

Response: Thank you for the suggestions, we have included the following limitations:

‘A further limitation is the low participation rate of 53%, which could lead to sampling bias. It is possible that adolescents with mental health problems and/or adolescents who had experienced several negative life events were less likely to participate in the survey. Thus, the prevalence estimates for depressive symptoms and negative life events could be underestimated in the current study, though it has been suggested that measures of association are less affected by selective participation (56). As the present study included adolescents from the general population, where only a small percentage had experienced several negative life events, it is a pertinent question whether the results would be different in a more vulnerable population. Further, only late adolescents were included in the study and the findings cannot be generalized to younger adolescents. It is further possible that positive life events may have a buffering effect on the association between negative life events and emotional distress (57), and it is possible that inclusion of positive life events could lead to more nuanced findings.

Conclusion - please remove the word “predicted” from lines 478 and 482, prediction is not possible with this cross-sectional data set.

Response: We have made the suggested alternations to the conclusion.

Minor Points

Unclear meaning of the sentence beginning on line 66 “positive development is in itself not sufficient to establish that resilience is present.” Please explain and provide references.

Response: We have included additional references and attempted to clarify the distinction between positive development and resilience as follows:

‘Positive development is in itself not sufficient to establish that resilience is present (15, 16); in addition, there must be current or past risk with a known potential to disrupt development (14). Thus, positive adjustment refers to an outcome of resilience, while resilience in itself is the process of overcoming risk (16).’

In might be helpful to provide further detail on gender differences in the experience of negative life events (lines 95-96).

Response: We have now rewritten the sentence and clarified that girls report a higher number of negative life events compared to boys: 

‘There are well-known gender differences in depressive symptoms (4, 6, 8, 28), and in the experience of negative life events (29), where girls report higher scores on both.’

Please insert stats for Family Cohesion (beta, p) in lines 313-314 to be consistent with remainder of results reported.

Response: Thank you for pointing this out, we have included the stats for Family Cohesion.

---

## [Decision Letter · Decision Letter 1]

20 May 2020

Life events and adolescent depressive symptoms: Protective factors associated with resilience

PONE-D-19-26146R1

Dear Dr. Askeland,

We are pleased to inform you that your manuscript has been judged scientifically suitable for publication and will be formally accepted for publication once it complies with all outstanding technical requirements.

With kind regards,

Kenji Hashimoto, PhD

Section Editor

PLOS ONE

Additional Editor Comments (optional):

Reviewers' comments:

Reviewer's Responses to Questions

**Comments to the Author**

1. If the authors have adequately addressed your comments raised in a previous round of review and you feel that this manuscript is now acceptable for publication, you may indicate that here to bypass the “Comments to the Author” section, enter your conflict of interest statement in the “Confidential to Editor” section, and submit your "Accept" recommendation.

Reviewer #1: All comments have been addressed

2. Is the manuscript technically sound, and do the data support the conclusions?

Reviewer #1: Yes

3. Has the statistical analysis been performed appropriately and rigorously? 

Reviewer #1: Yes

4. Have the authors made all data underlying the findings in their manuscript fully available?

Reviewer #1: Yes

5. Is the manuscript presented in an intelligible fashion and written in standard English?

Reviewer #1: Yes

6. Review Comments to the Author

Reviewer #1: (No Response)

7. PLOS authors have the option to publish the peer review history of their article (what does this mean?). If published, this will include your full peer review and any attached files.

Reviewer #1: No

---

## [Editor Report · Acceptance letter]

27 May 2020

PONE-D-19-26146R1 

Life events and adolescent depressive symptoms: Protective factors associated with resilience 

Dear Dr. Askeland:

I am pleased to inform you that your manuscript has been deemed suitable for publication in PLOS ONE. Congratulations! Your manuscript is now with our production department. 

With kind regards,

on behalf of

Prof. Kenji Hashimoto 

Section Editor

PLOS ONE